# Femtosecond time-resolved two-photon photoemission studies of ultrafast carrier relaxation in Cu$_2$O photoelectrodes

Mario Borgwardt [1], Stefan T. Omelchenko[2,3], Marco Favaro[1], Paul Plate[1], Christian Höhn[1], Daniel Abou-Ras[4], Klaus Schwarzburg[4], Roel van de Krol [1], Harry A. Atwater [2,3,5], Nathan S. Lewis [3,5,6], Rainer Eichberger[1] & Dennis Friedrich [1]

Cuprous oxide (Cu$_2$O) is a promising material for solar-driven water splitting to produce hydrogen. However, the relatively small accessible photovoltage limits the development of efficient Cu$_2$O based photocathodes. Here, femtosecond time-resolved two-photon photoemission spectroscopy has been used to probe the electronic structure and dynamics of photoexcited charge carriers at the Cu$_2$O surface as well as the interface between Cu$_2$O and a platinum (Pt) adlayer. By referencing ultrafast energy-resolved surface sensitive spectroscopy to bulk data we identify the full bulk to surface transport dynamics for excited electrons rapidly localized within an intrinsic deep continuous defect band ranging from the whole crystal volume to the surface. No evidence of bulk electrons reaching the surface at the conduction band level is found resulting into a substantial loss of their energy through ultrafast trapping. Our results uncover main factors limiting the energy conversion processes in Cu$_2$O and provide guidance for future material development.

[1] Institute for Solar Fuels, Helmholtz-Zentrum Berlin für Materialien und Energie GmbH, Hahn-Meitner-Platz 1, 14109 Berlin, Germany. [2] Division of Engineering and Applied Sciences, California Institute of Technology, Pasadena, CA 91125, USA. [3] The Joint Center for Artificial Photosynthesis, California Institute of Technology, Pasadena, CA 91125, USA. [4] Department Nanoscale Structures and Microscopic Analysis, Helmholtz-Zentrum Berlin für Materialien und Energie GmbH, Hahn-Meitner-Platz 1, 14109 Berlin, Germany. [5] Kavli Nanoscience Institute, California Institute of Technology, Pasadena, CA 91125, USA. [6] Division of Chemistry and Chemical Engineering, California Institute of Technology, Pasadena, CA 91125, USA. Correspondence and requests for materials should be addressed to R.E. (email: eichberger@helmholtz-berlin.de) or to D.F. (email: friedrich@helmholtz-berlin.de)

Photoelectrochemical (PEC) splitting of water into hydrogen and oxygen is a renewable method of hydrogen production that combines solar energy collection and water electrolysis into a single process[1]. In this process, sunlight absorbed by a semiconductor generates photoexcited charge carriers (electrons and holes), which drive the water-splitting (reduction or oxidation) reactions on the surface of the photoelectrode[2]. Solar-to-hydrogen efficiencies exceeding > 20% have been realized by use of high-quality III–V semiconductor photoelectrodes (e.g., GaAs, GaInP$_2$, etc.)[3–6], but the lack of long-term stability in aqueous electrolytes, as well as the materials scarcity and manufacturing cost of these photoelectrodes remain barriers to practical implementation of such systems. Metal oxide semiconductors provide potentially cheap and abundant candidate photoelectrodes[7–9]. Specifically, cuprous oxide (Cu$_2$O), a p-type semiconductor with a bandgap of $E_g$ = 1.9–2.2 eV, has a maximum theoretical water-splitting efficiency of ~18% under air mass (AM) 1.5 illumination[10–13]. Cu$_2$O photocathodes have shown high photocurrent densities (~8 mA/cm$^2$, which could potentially lead to efficiencies of ~10%) but exhibit lower-than-optimal photovoltages. Buffer layers, such as ZnO and Ga$_2$O$_3$, increase the photovoltage by tuning the junction potential and optimizing the conduction-band alignment at the Cu$_2$O/buffer layer interface[14–16]. To enhance the kinetics of the desired surface electrochemical reactions, such as hydrogen production or CO$_2$ reduction, noble metals such as Pt are usually deposited as co-catalysts either on the buffer layer or directly onto the surface of the photoelectrodes[11,17–19].

Although substantial progress has been made in optimizing Cu$_2$O photocathodes by using a p–n heterojunction strategy together with a buffer layer, beneficial advances in the fundamental understanding of the underlying processes would include elucidation of the carrier dynamics, charge separation and recombination in the bulk material, as well as across junctions, and additional information on the mechanisms by which these processes limit the overall energy conversion[20,21]. Accordingly, we describe herein the use of time-resolved two-photon photo-emission spectroscopy (tr-2PPE) to investigate under ultrahigh vacuum (UHV) conditions the electron dynamics and energetics at the surface of Cu$_2$O, as well as at the interface of Cu$_2$O with thin Pt adlayers. The method allows investigation of the energetic positions of occupied and unoccupied states, as well as the temporal evolution of transiently populated states.

2PPE spectroscopy has been previously successfully applied to study electron dynamics at surfaces and charge transfer processes across interfaces of metals and semiconductor model systems[22–24]. Only recently has its capabilities been utilized to study processes in emerging photovoltaic material systems, such as hot electron relaxation dynamics in hybrid metal–organic perovskite semiconductors[25–27]. We have now extended these efforts to the group of metal oxide semiconductors by including Cu$_2$O—one of the most promising metal oxide candidates for solar water splitting[28].

The dynamics of surface-trapped photogenerated electrons, as well as diffusion of photogenerated electrons from the bulk toward the surface, have been evaluated by comparing the spectral and dynamic signatures of reconstructed Cu$_2$O (100) single crystals before and after deposition of ultrathin adlayers of Pt. The results indicate that photoexcited electrons in Cu$_2$O lose a substantial fraction of their energy through ultrafast trapping in bulk defect states before arriving at the surface. The data also suggest that the Pt adlayer suppresses the slow accumulation of electrons into the surface defect states and that the Pt is capable of mediating charge transfer at the semiconductor/metal interface. Hence, our findings imply that the modest photovoltages that can be obtained from these Cu$_2$O samples primarily result from losses

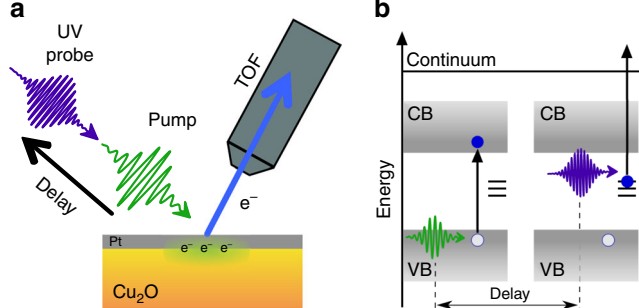

**Fig. 1** tr-2PPE working principle. **a** Schematic illustration of time-resolved two-photon photoemission spectroscopy (tr-2PPE) applied to Cu$_2$O utilizing a time-of-flight (TOF) spectrometer. **b** The transient PES study was performed with a pump laser intensity of 24 μJ/cm$^2$ at 494 nm wavelength (~2.5 eV energy), giving rise to resonant excitation above the bandgap. The electron population distribution among the transient states was probed by a 274 nm laser pulse (~4.5 eV energy) and was recorded as a function of pump–probe delay

derived from bulk recombination processes into bulk defect states.

## Results

**Time-resolved two-photon photoemission spectroscopy**. Time-resolved 2PPE is a pump–probe technique in which one of the beams is guided through an optical delay stage of variable length to generate an adjustable time delay between two femtosecond laser pulses (Fig. 1a). The laser pulse arriving first at the sample is used to photoexcite carriers to intermediate states (pump pulse), and the second pulse (probe pulse) promotes the photoexcited electrons to a final state above the vacuum level. The kinetic energy of these electrons provides direct information on the energy of the intermediate, normally unoccupied, states at the surface (Fig. 1b). The temporal evolution of the photoexcited electron distribution can moreover be evaluated by recording spectra as a function of the pump–probe delay. In the experimental configuration used herein, the temporal resolution was about ~35 fs (see Supplementary Fig. 1 for details). Artificially synthesized Cu$_2$O crystals were chosen, and an extensive surface reconstruction procedure was employed that ensures high reproducibility accounting for the surface sensitivity of the tr-2PPE technique. Investigations conducted at crystals without any surface treatment (as-received) yielded qualitatively similar results and are not shown.

The tr-2PPE study was performed with a pump pulse (494 nm, 24 μJ/cm$^2$) that induced a resonant excitation above the bandgap of Cu$_2$O. The electron population distribution among the transient states was probed by a subsequent laser pulse with a wavelength of 274 nm. The transient signal (TS) was derived by subtracting the photoelectron spectra recorded at negative delay times, representing a background spectrum, from the pump–probe spectra.

**Electron dynamics and electronic structure**. The color maps of Figs 2a, c show the two-dimensional dependence of the TS on the electron kinetic energy and on the pump–probe delay, for the reconstructed and Pt-covered Cu$_2$O samples, respectively. Representative spectra at selected pump–probe delays up to 1 ns are shown in Figs 2b, d, in which the bottom panels show the data after background subtraction. The steady-state energy structure (i.e., the occupied states) of the clean and Pt-deposited Cu$_2$O samples was inferred from spectra recorded using only the probe

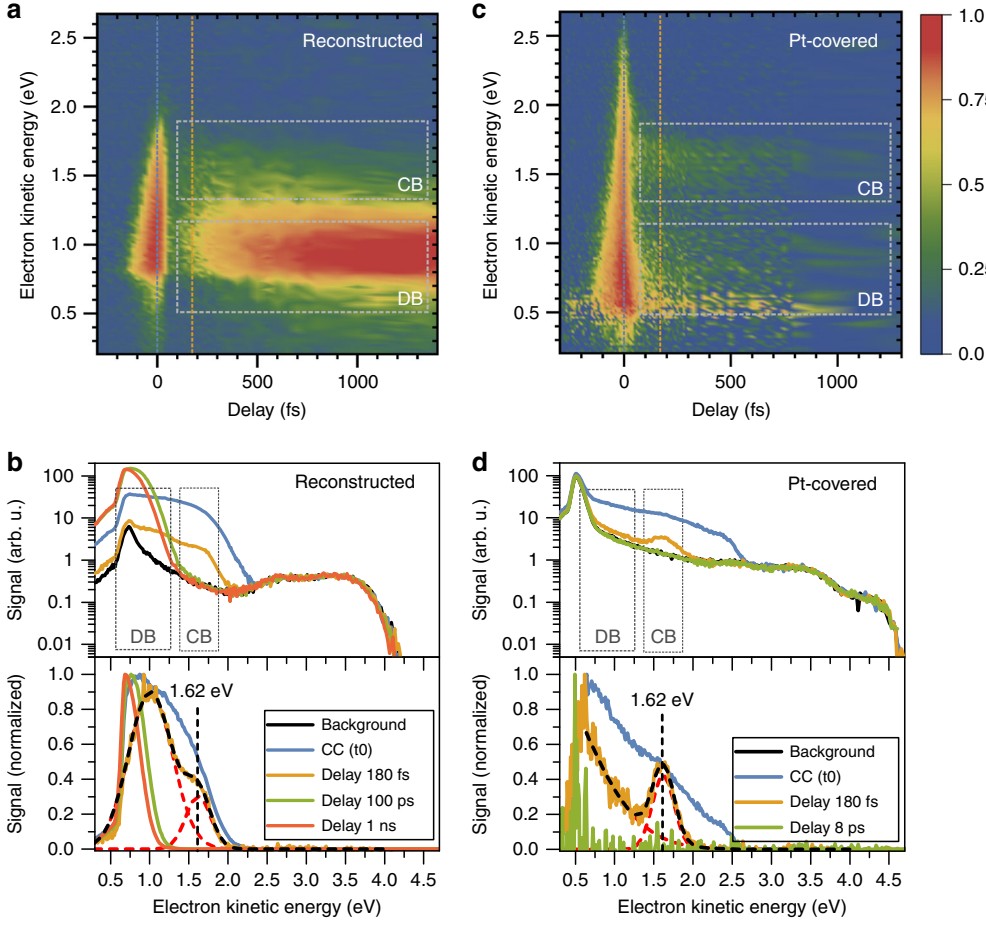

**Fig. 2** tr-2PPE spectra. Color maps of the transient photoemission signal as a function of the electron kinetic energy and the pump–probe time delay for the reconstructed $Cu_2O$ (100) surface (**a**), and for Pt-covered $Cu_2O$ (**c**). In **b** and **d**, selected spectra are shown at the specified pump–probe delays before (top) and after (bottom) background subtraction. In **a–d** the positions of the conduction band and defect band are indicated (CC: cross-correlation, DB: defect band, CB: conduction band). Source data are provided as a Source Data file

beam (see Supplementary Fig. 2). We point out that the recorded spectra consist of a combination from two different ionization processes involving different photoionization energies and originate from initially occupied and unoccupied states. For example, the stationary background (Fig. 2, black line) arises from a transition comprising two probe photons, whereas the actual 2PPE signal is composed of a mixed transition involving one pump and one probe photon (see Supplementary Fig. 3). Hence, the definition of a single binding energy scale representing all signals is not feasible and the directly measured electron kinetic energy scale was chosen. However, each spectral component can be referenced to the system's internal Fermi energy level with $E_F = 0$ and the calibration of the binding energy scale was performed by measuring the Fermi edge of a Cu reference sample (see Supplementary Fig. 4 and Supplementary Note 1). The valence-band maximum (VBM) was found to be 0.5 eV below the Fermi energy of the reconstructed $Cu_2O$ surface. The Pt-covered $Cu_2O$ sample exhibited a similar steady-state energy structure to that of the reconstructed $Cu_2O$ surface, except that the Pt adlayer provided an extension of the signal up to the Fermi level, providing a clear signature of the metallic character of the adlayer. XPS, as well as SEM measurements were indicative of a non-conformal structure of the Pt adlayer (see Supplementary Methods and Supplementary Figs. 5–10 for details). The island-like growth of the Pt adlayer is consistent with a non-uniform distribution of pinholes at the reconstructed surface.

For low photoelectron kinetic energies, the inelastic mean-free paths are relatively large, exceeding tens of nanometers[29]. However, because elastic electron-acoustic phonon scattering overcomes the energy loss scattering events, the photoelectron escape depth is substantially reduced for such low kinetic energies[30,31]. Hence, the thickness region sampled by 2PPE is typically on the order of a few nanometers[25,32,33]. In addition to the above-mentioned surface sensitivity of 2PPE, it is important to note that due to the non-conformal coverage of the Pt adlayer and the surface area accessible by the probe pulse with a diameter of ~50 μm, both Pt-covered and uncovered $Cu_2O$ areas were simultaneously probed. We therefore expect that the $Cu_2O$-derived features on the Pt-covered sample primarily originate from a combination of directly emitted electrons from the $Cu_2O$ substrate in conjunction with electrons probed through the Pt top layer.

The transient features (unoccupied states) can be inferred from the color maps, which showed several distinct features for both samples. In the vicinity of zero-time delay (± 100 fs), the TS exhibited a strong signal at kinetic energies between 0.5 and 2.5 eV. This feature showed a sharp trailing edge at the end of the pump pulse (~100 fs) and exhibited an increase in asymmetry, as the kinetic energies decreased and as the time delay became increasingly negative. The symmetric part of this feature can be consistently attributed to the cross-correlation (CC) signal originating from the sample, and is limited by the instrument response. The asymmetric part of the signal at negative time

delays can be consistently assigned to thermal cooling of an initially hot electron distribution that relaxed in ~50 fs toward the conduction-band minimum (further details in Supplementary Fig. 11).

For the Pt-covered sample, a resonant feature centered at ~1.62 eV kinetic energy corresponded to a binding energy of −1.65 eV, and disappeared within 1 ps. The same feature was also present in the signal of the reconstructed $Cu_2O$ sample, but in that case, the signal was difficult to resolve unambiguously, due to an adjacent intense signal at lower kinetic energies that partially overlapped with the resonance of interest. In both cases, the energy difference between the resonance of interest and the position of the VBM was 2.15 eV, in excellent accordance with the expected electronic bandgap energy of $Cu_2O$. This resonance can thus consistently be assigned to the conduction-band (CB) level. The appearance of this characteristic feature on both samples (reconstructed and Pt covered) strongly indicates that electrons originating from $Cu_2O$ can be probed on both samples, and provides a clear basis for understanding the semiconductor band alignment.

In the reconstructed sample, an intense, long-lived signal at energies of ~1 eV was observed ~200 fs after the initial pump pulse (see Fig. 2a), with the signal shifting to lower kinetic energies at time delays larger than 1 ps. This spectral region encompasses states located within the bandgap, and these signals are therefore likely to originate from defects.

**Origin of defect levels by bulk-sensitive photoluminescence**. In order to gain additional information about the origin of the filling of these defect levels located within the bandgap, photoluminescence (PL) measurements were conducted exhibiting primarily bulk sensitivity and, therefore, represent a well-suited complementary method to the surface-sensitive 2PPE technique. The experiments were conducted at the same stages of sample preparation (reconstructed, Pt deposited) without breaking UHV conditions and applying identical pump laser pulse conditions as utilized in the 2PPE measurements (Supplementary Fig. 12 and Supplementary Methods). Both stationary and time-resolved photoluminescence measurements (tr-PL) strongly suggest that Cu vacancies act as the dominant defect type in the investigated material, and the photoemission energy is in close accordance with the energetic position of the defect state in the bandgap. The different surface treatments did not alter the PL signals, suggesting a relatively high concentration of Cu vacancies in the bulk, and that the surface contribution, as well as influence to the charge recombination is negligible. Therefore, additional surface-sensitive experiments (LEED, XPS) were performed to obtain information about the stoichiometry, structure, and energetics (band bending, band alignment) of the material (see Supplementary Figs. 5–7 and Supplementary Discussion and the "Methods" section).

**Pt deposition reduces the 2PPE signal from defect states**. Surface analysis of the reconstructed and Pt-covered $Cu_2O$ sample by means of atomic-force and scanning electron microscopy (AFM, SEM; see Supplementary Figs. 8–10 and Supplementary Methods) revealed a non-uniform defect distribution; hence, the long-lived signal can consistently be ascribed to sites associated with a high dislocation density. When the defect band intensity was normalized to the amplitude of the conduction-band signal (bottom panel of Fig. 2b, d), the 2PPE signal from the defect states in the reconstructed $Cu_2O$ was much more intense than the analogous signal exhibited by the Pt-covered $Cu_2O$ sample. This behavior suggests that either the defect states disappeared after deposition of Pt, or that the Pt provided an

alternative pathway for the photoexcited electrons that prevented accumulation of charge carriers in the defect states.

**Temporal evolution of electron density at the interfaces**. Two main spectral regions were integrated to facilitate comparison of the temporal dependence of the electron concentration at both interfaces. The low-energy region (0.5–1.3 eV) corresponds to the defect states, whereas the high-energy region (1.7–2.5 eV) corresponds to the CB (Fig. 2c). Figure 3 shows the data normalized to the CC peak at zero-time delay. The most substantial difference between both samples was observed in the low-energy region, which corresponds to the defect states (orange curves). The low-energy electron yield of the reconstructed surface exhibited a pronounced increase vs. time, whereas the electron signal only decayed on Pt-covered $Cu_2O$ surfaces. At ~200 fs after the pump pulse excitation, the initial occupation level for the reconstructed surface increased by more than one order of magnitude and extended into the nanosecond domain (not shown here, for extended transients, see Supplementary Fig. 13). This substantial increase in electron density is consistent with a diffusion process of photoexcited electrons from the bulk to the surface. Similar findings have been reported for tr-2PPE studies on GaAs (100) surfaces, where a rising electron population on a ps timescale has been assigned to scattering of electrons into low-energy states in the band-bending region[34]. The considerable temporal delay after the excitation pulse supports this assignment, in that the majority of electrons detected at the defect level were not initially created at the surface. The electron yield rise was fitted with a biexponential model, yielding a rapid time constant on the order of 1 ps and a second, slower component in the range of 80 ps. Intensity-dependent measurements (Supplementary Fig. 13) revealed that the time constant of the fast-rise component of the signal decreased with increasing pump pulse intensity, whereas the second component of the signal was independent of the pump/pulse intensity. The Pt-covered sample exhibited no change in 2PPE transients over the same range of pump powers.

The integrated signal from the conduction-band electrons (1.7–2.5 eV, blue curves in Fig. 3) exhibited identical behavior for both the Pt-coated and reconstructed $Cu_2O$ samples. The initial CC peak was followed by a single exponential decay, with a decay constant of $110 \pm 10$ fs in both cases. Thus, surface electrons initially generated in the CBM relax on this ultrafast timescale into lower-lying defect states. No evidence was obtained for a subsequent diffusion process of charge carriers that were excited in the bulk reaching the surface at the CB level.

**Discussion**

Time-resolved 2PPE spectroscopy provides detailed insight into the ultrafast charge-carrier dynamics and relaxation processes at the surface of reconstructed $Cu_2O$ (100) electrodes, as well as at $Cu_2O$ (100)/Pt interfaces. The energetic data indicate that downward band bending[35] (Fig. 4a) in ultrahigh vacuum provides a driving force for electrons to move to the $Cu_2O$ surface, where they contribute to the 2PPE signals (compare Fig. 2). Two distinct photoemission regions were observed for both samples, ascribable to conduction-band states at higher kinetic energy and mid-gap states at lower kinetic energy, respectively. Due to the high pump photon energy compared with the optical bandgap, we expect to generate initially free electrons in the CB followed by subsequent formation of excitons being well known to be present in $Cu_2O$ and exhibiting large exciton-binding energies of about 150 meV[13]. The relaxation of free electrons into excitons would lead to a spectral feature slightly below the initial level, although we point out that both contributions would be difficult to distinguish. However, at CB levels ascribed to either species, the signal was relatively weak, and disappeared within

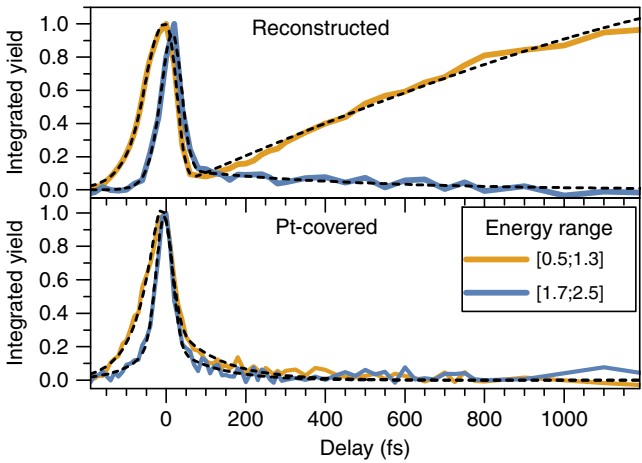

**Fig. 3** Integrated electron yield of the reconstructed (top) and Pt-deposited (bottom) samples for two different spectral regions. Fits (black, dashed) were obtained by using exponential decay or biexponential rise models convoluted with a symmetric Gaussian shape to account for the instrument response. The asymmetry at negative time delays resulting from the temporal evolution of an initially hot electron distribution relaxing toward the conduction band minimum was added to the fit. Source data are provided as a Source Data file

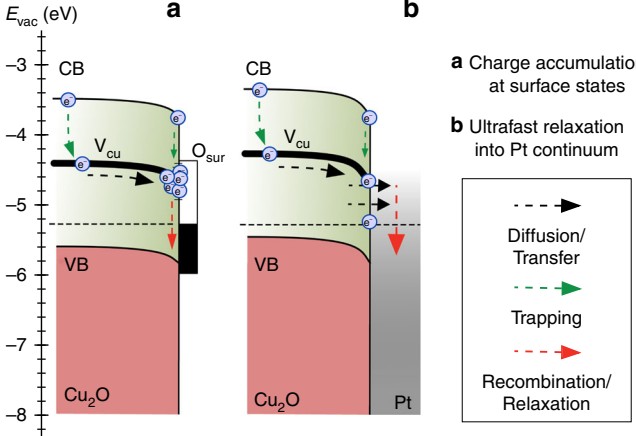

**Fig. 4** Energy-band diagram. Band bending and carrier dynamics at the surface of the reconstructed (**a**) and Pt-deposited $Cu_2O$ (100) single crystals (**b**). The energy-band positions and band bending were measured by steady-state 2PPE and XPS measurements referenced to the vacuum level. In **a**, charge transfer and accumulation occur at defect states ($V_{Cu}$) located within the bandgap. In contrast, in **b**, deposition of a Pt adlayer leads to ultrafast charge extraction and relaxation into the Pt continuum

the first ps after excitation (Fig. 3), consistent with ultrafast capture of conduction-band electrons or excitons into the broad mid-gap band presumably originating from Cu vacancies[36–38] (dotted green arrows in Fig. 4).

The reconstructed sample exhibited a much higher photoelectron yield at the mid-gap levels, consistent with carriers from these defects populating a large density of long-lived surface acceptor states. The long rise time of the integrated 2PPE signal, which extended into the ns time domain, is consistent with these states being filled with bulk electrons that drift or diffuse toward the surface with the electrons then primarily accumulated at sites

with a high dislocation density. The width of the space-charge layer is only ~8 nm (Supplementary Discussion, Surface band bending), and most of the electrons are slow to arrive at the surface, as evidenced by the PL results, consistent with most CB electrons being trapped in the putative Cu vacancy states well before the electrons reach the surface. This fits well to the low electron mobility estimated by Paracchino et al.[39] between 2.7 and 6.3 $cm^2\,V^{-1}\,s^{-1}$ in electrodeposited $Cu_2O$ using time-resolved terahertz spectroscopy. Considering the position of the $V_{Cu}$ defect band located deep below the conduction band minimum, bulk electron transport to the surface would consistently occur via thermally assisted hopping within the defect band, in accordance with the moderate values for minority-carrier mobility and diffusion length from other reports for $Cu_2O$[11,40–42]. Assuming that surface accumulation of the electrons occurs at levels isoenergetic with the bulk trap band implies the presence of a continuous defect band up to the crystal surface.

The mid-gap level for the Pt-covered sample (Figs 2, 4b and bottom of Fig. 3) exhibited weak photoemission signals ascribable to defect states directly populated by the optical pump pulse. Deposition of the noble metal adlayer moreover suppressed the delayed electron signal rise. The surface treatment did not however substantially affect the measured band bending (Supplementary Discussion, Surface band bending). Accordingly, the transport of bulk electrons toward the surface is not changed due to Pt deposition. These data are consistent with the electrons being rapidly transferred into the continuum of Pt states upon arrival at the $Cu_2O$/Pt interface. The carriers would then promptly relax toward the system Fermi level, where they would not be detected experimentally due to the energy cutoff of the 2PPE setup for low kinetic energies.

Noble metal co-catalysts are often deposited onto the surface of photoabsorbers to enhance water splitting or $CO_2$ reduction[17,43]. Pt is a material of choice for such processes and the Pt can be deposited either directly onto the semiconductor surface or onto an intermediate buffer layer[11,18,43,44]. Chatchai et al.[45] deposited Pt on $Cu_2O$ for hydrogen evolution and observed higher photocatalytic activity for decorated vs. bare $Cu_2O$ surfaces. To the extent that the surfaces in UHV provide useful comparisons to those in contact with an electrolyte, the observations herein suggest that the Pt may not primarily act as a passivation layer that inhibits recombination, but instead Pt primarily can mediate ultrafast carrier transfer across the internal interface by direct coupling to the $Cu_2O$ defect band.

At voltages much closer to the intrinsic bandgap energy of the material, only electrons at higher energetic levels presumably contribute to the photocurrent[46–48]. However, the intrinsic semiconductor defect band produces substantial open-circuit voltage losses in $Cu_2O$. Hence, in addition to known issues in $Cu_2O$, such as a mismatch between the electronic band alignment and defect states at heterojunction interfaces, bulk defect states in the $Cu_2O$ may substantially limit the obtainable voltage in $Cu_2O$ devices. These findings provide insight into materials with high trap densities that produce high photocurrents, but that have photovoltage losses that are more difficult to identify. Different preparation methods reported for $Cu_2O$ result either in photocathodes that exhibit high photovoltage[49] or high photocurrent densities, but only recently have high-quality, thermally oxidized $Cu_2O$ layers exhibited simultaneous improvement in both quantities[40,50]. To evaluate the potential of a material for photovoltaic applications or subsequent chemical reactions, a surface-sensitive technique with high time resolution is thus beneficial to facilitate determination of the energetic distribution of charge carriers in the light absorber. Our results suggest that 2PPE complemented by PL spectroscopy is a useful method to

characterize various materials and material synthesis methods, for photovoltaic and photoelectrochemical applications.

In conclusion, before photoexcited electrons arrive at the surface, in $Cu_2O$, these electrons lose a substantial fraction of their energy through ultrafast trapping into bulk defect states. No evidence was found for bulk electrons reaching the surface at CB energy levels. The spectroscopy implies that the modest photovoltages that can be obtained from these $Cu_2O$ samples are not primarily due to recombination induced by surface states, but result from losses derived from bulk recombination processes into bulk defect states. For reconstructed $Cu_2O$ (100) surfaces, the electrons accumulated in long-lived (sub)surface states upon arrival at the surface. In contrast, a Pt adlayer mediated ultrafast extraction of electrons and made them available for subsequent photochemical conversion steps. This process increased the photocurrent, but the resultant energy loss upon transfer of the electron to the Pt further reduced the accessible photovoltage. The observations indicate that tr-2PPE can provide a powerful experimental method to simultaneously unravel the energetics and dynamics of photoexcited electrons as they arrive at surfaces or internal interfaces of semiconductors. Tr-2PPE thus provides explicit insight into the origin and mechanism of photovoltage losses in photovoltaic or photocatalytic materials and can offer crucial insights at an early stage of materials development.

## Methods

**Time-resolved two-photon photoemission spectroscopy**. Laser pulses were generated with a pulsed high-repetition-rate (150 kHz) Ti:sapphire laser system with two low-power non-colinear optical parametric amplifiers (NOPAs), that provided light in the visible spectrum. Subsequent second-harmonic generation of the output of one NOPA provided UV photons for the probe pulse. The time delay between both pulse trains was produced using an electronically controlled delay stage to vary the optical path of one of the beams. The probe beam photon energy was maintained at $hv_{probe} = 4.5$ eV, and the pump photon energy was set at $hv_{pump} = 2.5$ eV. Satisfactory time resolution in the 2PPE experiments was obtained by compressing the pulses with prism pairs, which resulted in a cross-/auto correlation of <50 fs. The kinetic energy of the photoemitted electrons was measured with a homemade time-of-flight spectrometer.

**$Cu_2O$ sample preparation**. Cuprous oxide single crystals were grown by the following method: feed and seed rods were grown by the thermal oxidation of high-purity Cu rods (Alfa Aesar, 99.999%) in a vertical tube furnace (Crystal Systems Inc.) in air for 100 h at 1050 °C. The rods were then cooled in $N_2$ at 120 °C/h. Prior to growth, the rods were cleaned in acetone and etched using dilute nitric acid (0.1 M) for 60 s. The rods were suspended by either Cu or Pt wire. Single crystals were grown in an optical floating zone furnace (CSI FZ-T-4000-H-VII-VPO-PC). Crystals were grown in air with the seed and feed rods counter-rotating at 7 rpm. Single crystallinity was confirmed using X-ray diffractometry and pole figure analysis. The resulting single-crystalline boules were cut to the (100) plane and polished (SurfaceNet GmbH). Surface reconstruction of the $Cu_2O$ (100) single crystal was achieved by 3–4 cycles of $Ar^+$ ion bombardment (800 V, 1 μA, 50 min) and annealing at 550 °C for 30 min in UHV.

**Photoelectron spectroscopy**. X-ray photoelectron spectroscopic (XPS) measurements were performed under ultrahigh vacuum using an Al Kα X-ray source (1486.74 eV) equipped with a SPEC FOCUS 500 monochromator. A hemispherical XPS analyzer supplied by SPECS (Phoibos 100) was used with a source-to-analyzer angle of 54°. Pass energies of 30 and 10 eV and scan steps of 0.5 and 0.05 eV were used to obtain survey and fine spectra, respectively.

Ultraviolet photoelectron spectroscopic (UPS) data were obtained using a helium-discharge lamp emitting He Iα (21.22 eV) radiation. The ground of the sample holder was used to bias the sample. The analyzer equipment was the same as that used for XPS measurements, and UPS data were collected using the same parameters as those for XPS fine spectra.

All data were obtained at room temperature in ultrahigh vacuum (UHV) conditions (base pressure below $1 \times 10^{-10}$ torr).

## Data availability

The source data underlying Figs. 2a–d and 3 are available in Zenodo, https://doi.org/10.5281/zenodo.2628238. The source data for the Supplementary Figures are available from the corresponding author upon reasonable request.

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

## Acknowledgements

M.B., R.E. and D.F. thank P. Sippel for discussions. M.B. acknowledges funding from the Helmholtz Association through the Excellence network UniSysCat (ExNet-0024-1). This work was supported in part (S.T.O., H.A.A. and N.S.L.) through the Office of Science of the U.S. Department of Energy (DOE) under award no. DE-SC0004993 to the Joint Center for Artificial Photosynthesis, a DOE Energy Innovation Hub. D.F. acknowledges support by the German Research Foundation (DFG), project numbers PAK 981/1 and FR 4025/2-1.

## Author contributions

M.B., D.F. and R.E. designed the experiments on samples provided by S.T.O, H.A.A., and N.S.L. M.B. and D.F. carried out the laser experiments and analyzed the data with the help of R. vdK and R.E. M.B., M.F., P.P., C.H. and D.F. performed the LEED and XPS measurements and analyzed the data. D.A.-R. performed the SEM/EDX experiments. K. S. performed the AFM measurements. M.B., R.E. and D.F. prepared the paper. All authors discussed the results and commented on the paper.

## Additional information

**Competing interests:** The authors declare no competing interests.

