## [Peer Review File · Nature Communications]

Reviewers' comments:

Reviewer #1 (Remarks to the Author):

The manuscript here presents time-resolved two-photon photoemission studies of Cu₂O photoelectrons. The manuscript is interesting. However, I was confused by the overall message. On the one hand the authors attempt to motivate the work to demonstrate that the technique is useful for studying the surface carrier dynamics of photoelectrons. In that regard they determine the effect of the carrier dynamics between a bare Cu₂O surface and one with a Pt overlayer catalyst. On the other hand the authors discuss the importance of Cu₂O. An present the work to understand the carrier dynamics at the surface of this important semiconductor. Since there are significant differences in the response for the bare and Pt overlay sample this is something to learn about how these photoelectrons are working. But I didn't get a clean message as to what the Pt is doing. Thus, the biggest issue with this manuscript is that there is no insight for how one might develop a Cu₂O photoelectrode that is not impacted by the defects. Furthermore, the measurements are done in ultrahigh vacuum - which seems to lessen the impact of the measurements and the measurement technique somewhat. While the authors do make an attempt to justify why these measurements are still valid.

The discussion and presentation of Fig. 2 is confusing. What determines the zero energy scale here? Is the Fermi energy not determined by the system for these measurements? Why is the CB at a lower energy than the VBM and shouldn't the Fermi energy be between the CB and VBM. I see that the DB is at a lower energy than the CB which would make sense. But it's unclear from this plot how the energy diagrams of Fig. 4 are constructed. Maybe this is well known for the tr-2PPE community (but this is probably small).

Could the authors change the excitation wavelength. This would help determine if the rise time in Fig. 3 is due to diffusion of carriers within the bulk to the surface. How do the authors determine the CB electrons first relax to DB and then transport to the surface. Couldn't CB diffuse to the surface and then relax to the DB? The authors rule this out due to the fast decay of surface CB to the DB and then the subsequent arrival of DB electron density. However, if the relaxation of surface CB electron to the DB is fast then one might expect a similar behavior. As CB electrons arrive at the surface (in coherently) they quickly relax to the DB.

The main conclusion is that photoelectrons relax to bulk defect states prior to reaching surfaces and thus lose their potential very fast interior to the material. Does this mean that Cu₂O cannot be fixed? If the defects are due to Cu vacancies is there a way to decrease their number and see how that impacts the results.

Overall. I like the manuscript - but not sure that this should be published in Nature communication. The impact of the manuscript seems rather low. Maybe the authors should use the results to improve the operation of the photoelectrons.

Reviewer #2 (Remarks to the Author):

In the manuscript 'Femtosecond time-resolved two-photon photoemission studies of ultrafast carrier relaxation in Cu₂O photoelectrodes' M. Borgwardt et al. report on a time-resolved 2PPE study of the carrier dynamics in a pristine Cu₂O single crystal and after deposition of a small amount of Pt. From their data they conclude on Cu bulk vacancy states acting as decay channels for charge carriers

photoexcited into the conduction band and being responsible for the reduced photovoltages observed for this material system. The experiments have been performed thoroughly and the data were analyzed carefully. The main text itself misses sometimes relevant details, which are partly 'hidden' in the supplementary information. The interpretation stays at qualitative level but is convincing at least regarding the major conclusions.

As the key outcome of the study the authors conclude that the 'modest photovoltages that can be obtained from these Cu₂O samples are not primarily due to recombination induced by surface states, but primarily from bulk recombination processes into bulk defect states.' This seems to me an interesting finding, but I am not really convinced that this is enough to justify publication in Nature Communications. I have to admit, however, that experts in the field of solar-driven water splitting processes may come to a different conclusion.

In case the manuscript is considered for publication in Nature communications, the following points should be answered by the authors and considered in a revised version:

(1) The authors state in their conclusions that their 'observations indicate that tr-2PPE can provide a powerful experimental method to simultaneously unravel the energetics and dynamics of photoexcited electrons as they arrive at surfaces or internal interfaces of semiconductors. Tr-2PPE thus provides explicit insight into the origin and mechanism of photovoltage losses in photovoltaic or photocatalytic materials and can offer crucial insights at an early stage of material development.' This is definitely correct, but in my opinion not actually new or surprising. Tr-2PPE experiments have been performed over more than 30 years and a multitude of experiments on quite different material systems from many groups were published in the past, also including experiments addressing relevant processes in photovoltaic systems. I actually was very surprised to see that the authors refer only to a very limited number of tr2PPE studies (in the supplemental information, only). Even more, these citations are in my opinion not really of relevance for the key outcome of the study. I strongly recommend checking for instance tr2PPE studies by the Zhu group on inorganic-organic perovskites. As far as I know also the Perfetti group and the Fauster group (D. Niesner) have published some work on this stuff. In a wider context also work by the Wolf group on charge carrier solvation in ice overlayers and work by the Harris group and the Höfer group on charge-transfer processes across interfaces could be of interest. Finally, I remember some work by the Aeschlimann group on surface charge accumulation in GaAs due to carrier diffusion from the bulk. I am pretty sure that at least part of this work could be helpful in the interpretation of the submitted study.

(2) It becomes not clear what amount of Pt was finally deposited onto the reconstructed Cu₂O surface that was investigated by tr2PPE. Is the surface fully covered with Pt or are there open Cu₂O areas in between the Pt islands that form upon deposition as verified by AFM and SEM. In the main text, the authors refer to a non-conformal structure of the Pt adlayer, which may hint to open Cu₂O areas. However, in the supplementary information (section 6) the authors write: 'X-ray photoelectron spectroscopy (XPS) data for Pt films of increasing thickness revealed a non-conformal coverage for the thinnest films (0.5 nm) while a continuous Pt layer was obtained for longer deposition times (Fig. S8b). Based on these results, 1 nm Pt films were used for all of experiments discussed herein.' What does this finally mean for the samples discussed in the main text? The information is in my opinion important as the presence of open Cu₂O areas could be of relevance for the interpretation of the data.

(3) The authors directly conclude from the bulk-sensitive photoluminescence measurements onto the origin of the dominating signal recorded using the surface sensitive 2PPE method. I would suggest being more conservative with such statements as it is well known that the surface electronic structure can differ significantly from the bulk electronic structure as the bulk termination can result in a

surface reconstruction and also the formation of quite different types of defects.

(4) In line 112 the authors state that the Cu₂O-derived features on the Pt covered sample primarily originate from a combination of directly emitted electrons from the Cu₂O substrate (I assume this refers to Cu₂O areas not covered by Pt) in conjunction with electrons probed through the Pt top layer. I have to admit that I do not completely understand how the authors come to this conclusion from their discussion on the electron mean free path in the same paragraph. Particularly I am wondering how the authors can exclude that the Cu₂O-derived features on the Pt covered sample exclusively arise from directly emitted electrons from the Cu₂O substrate?

(5) The conclusion that the characteristic signal rise in the band-gap feature arises from carrier diffusion to the surface in the defect band seems to me very reasonable. I was surprised that the observed timescales were not related to the time-resolved PL data on the (much longer) defect depopulation times. Furthermore, the authors observe and mention in the main text a two-component signal rise with a pronounced fluence dependence of the fast rise-time. What can we learn from these observations? This point is not discussed at all in the rest of the manuscript.

(6) What mechanism channels excited electrons at the surface into sites associated with a high dislocation density (line 214)? Shouldn't such type of process result in the formation of an in plane surface potential gradient which at some point will stop the channeling process.

(7) The author mention thermally assisted hopping within the VCu defect band. What energies are we talking about? Here, experiments performed at different temperatures would be nice and would strongly support the interpretation in terms of bulk to surface diffusion.

(8) In the abstract the authors claim that their data suggest that the Pt adlayer reduced the surface defect states. Later in the manuscript (line 147) they write that '...either the defect states disappeared after deposition of Pt, or that the Pt provided an alternative pathway for the photoexcited electrons that prevented filling of defect states.' From the following discussion in the manuscript I learned that none of these statements is finally correct: The defects did not disappear but become hidden underneath the Pt-island. These defect states become still populated (at least as implied by Fig. 4(b)). However, in the presence of the Pt overlayer they get much faster depopulated in comparison to the reconstructed surface due to coupling to platinum electronic states.

(9) The authors state that the reconstructed surface has a non-uniform defect distribution (line 143) (and conclude from this that the 'long-lived signal can consistently be ascribed to sites associated with a high dislocation density.') I missed somehow, where this information is coming from. Maybe this point is somewhere hidden in the supplementary information. In any case at least a reference must be given.

(10) It seems to be very important that a reconstructed Cu₂O surface instead of a non-reconstructed surface is used in the experiment as this point is mentioned several times throughout the manuscript. I am not familiar with the details of the water-splitting capabilities of Cu₂O and I think it would be helpful for the non-expert to spend a few words in the main text or the supplementary information on this point.

Reviewer #3 (Remarks to the Author):

This work measures the relaxation dynamics of photoexcited electrons in Cu₂O pure and with a Pt

cover. The performed 2-photo-photoelectron measurements are well designed and executed. The manuscript describes the undertaken experiments and analyses clearly. I also summarizes the results clearly. The difference between pure and Pt-covered material is strikingly different and I think that the authors are right with their analysis. The defect sites on the surface become slowly filled with bulk electrons. it is quite fascinating to see this in real time.

I only have one very minor comment: in the abstract the authors write about energetically "unfavorable" states. It is not upfront clear what that should mean. The authors may want to chose a more clarifying description.

Otherwise, publication of this work is recommended.

Clemens Burda

Point-by-point response to the referees' comments:

Reviewer #1 (Remarks to the Author):

The manuscript here presents time-resolved two-photon photoemission studies of Cu₂O photoelectrons. The manuscript is interesting. However, I was confused by the overall message. On the one hand the authors attempt to motivate the work to demonstrate that the technique is useful for studying the surface carrier dynamics of photoelectrons. In that regard they determine the effect of the carrier dynamics between a bare Cu₂O surface and one with a Pt overlayer catalyst. On the other hand the authors discuss the importance of Cu₂O. An present the work to understand the carrier dynamics at the surface of this important semiconductor. Since there are significant differences in the response for the bare and Pt overlay sample this is something to learn about how these photoelectrons are working. But I didn't get a clean message as to what the Pt is doing. Thus, the biggest issue with this manuscript is that there is no insight for how one might develop a Cu₂O photoelectrode that is not impacted by the defects

We thank the reviewer for carefully reading and evaluating the manuscript. We agree with the reviewer's opinion in that the manuscript needs a clearer focus on its central outcome. We have adapted the manuscript accordingly (see changes to abstract below and in response to the comments hereinafter) and underline in more detail the importance of photovoltage losses in Cu₂O electrodes originating from deep defect states. However, we note that the manuscript was not intended to investigate specific aspects of material engineering, but rather unraveling limiting factors and providing deeper insight into this loss mechanism. We therefore believe that our results will be particularly helpful in future Cu₂O material development broadening the emphasis of current research from interface engineering towards a more profound and controlled synthesis of high-quality bulk material.

Associated changes to the abstract:

"By referencing ultrafast energy resolved surface sensitive spectroscopy to bulk data we identify the full bulk to surface transport dynamics for excited electrons rapidly localized within an intrinsic deep continuous defect band ranging from the whole crystal volume to the surface. No evidence of bulk electrons reaching the surface at the conduction band level is found resulting into a substantial loss of their energy through ultrafast trapping."

Furthermore, the measurements are done in ultrahigh vacuum - which seems to lessen the impact of the measurements and the measurement technique somewhat. While the authors do make an attempt to justify why these measurements are still valid.

As cross check, all tr-2PPE, PL and tr-PL measurements were also conducted directly after crystal cleavage without further surface treatment (as-received) resulting into qualitatively similar signals compared to the reconstructed samples demonstrating the general insensitivity to specific surface preparation methods. However, signal intensities

and background contributions with tr-2PPE were different and varied depending on respective sample and chosen spot on the sample surface. Such variations are not surprising due to the high surface sensitivity, so that we decided to include only the results from the well-defined and highly reproducible surface that were not hampered by these difficulties. On the other hand, our results suggest that even in contact with a solvent the main loss channel is ascribed to the decay of charge carriers into deep bulk defect states and thus would not alter the final conclusions.

Associated changes to the main text:

Page 4:

“Artificially synthesized Cu_2O crystals were chosen and an extensive surface reconstruction procedure was employed that ensures high reproducibility accounting for the surface sensitivity of the tr-2PPE technique. Investigations conducted at crystals without any surface treatment (as-received) yielded qualitatively similar results and are not shown.”

The discussion and presentation of Fig. 2 is confusing. What determines the zero energy scale here? Is the Fermi energy not determined by the system for these measurements? Why is the CB at a lower energy than the VBM and shouldn't the Fermi energy be between the CB and VBM. I see that the DB is at a lower energy than the CB which would make sense. But it's unclear from this plot how the energy diagrams of Fig. 4 are constructed. Maybe this is well known for the tr-2PPE community (but this is probably small).

We agree that the discussion of the used energy scale in Fig. 2 needs clarification. The photoemission spectra are presented in dependence on the directly measured electron kinetic energy. It is important to note, that the spectra can be understood as a superposition of two different ionization processes – one involving two probe photons (equally to solely applying the probe beam; black background curve) and, secondly, a mixed transition involving one pump and one probe photon. The transformation of the kinetic energy scale into a single common binding energy scale is therefore not achievable since the different combinations of pump and probe photons result into different binding energy scales. For the same reason the VBM originating from two probe photons occurs at higher absolute value on the electron kinetic energy scale than the CB and DB originating from a mixed transition involving one pump and one probe photon. For further clarification we refer to section 1 in the original version of the SI.

Associated changes to the main text:

Page 6:

“We point out that the recorded spectra consist of a combination from two different ionization processes involving different photoionization energies and originate from initially occupied and unoccupied states. For example, the stationary background (Fig.

2, black line) arises from a transition comprising two probe photons, whereas the actual 2PPE signal is composed of a mixed transition involving one pump and one probe photon. Hence, the definition of a single binding energy scale representing all signals is not feasible and the directly measured electron kinetic energy scale was chosen. However, each spectral component can be referenced to the systems internal Fermi energy with $E_F=0$ and the calibration of the binding energy scale was performed by measurement of the Fermi edge of a Cu reference sample (see Supplementary section 1).“

We further adapted Fig. 2 to avoid confusion and to simplify the general presentation of the measured results. We deleted the indicator for the VBM and E_F .

Could the authors change the excitation wavelength. This would help determine if the rise time in Fig. 3 is due to diffusion of carriers within the bulk to the surface. How do the authors determine the CB electrons first relax to DB and then transport to the surface . Couldn't CB diffuse to the surface and then relax to the DB? The authors rule this out due to the fast decay of surface CB to the DB and then the subsequent arrival of DB electron density. However, if the relaxation of surface CB electron to the DB is fast then one might expect a similar behavior. As CB electrons arrive at the surface (in coherently) they quickly relax to the DB.

The used frequency conversion of the Ti:Sapphire laser system via non-collinear optical parametric amplifiers (NOPA) is highly limited in the spectral range that can be achieved with reasonable pulse energy output. The limited range would result only into small variations in absorptions depth that might be difficult to compare on a qualitative level. We further point to the fact that the dynamics of 2PPE signal is particularly sensitive to the charge carrier dynamics in the space-charge region that might considerably deviate from the bulk. A fluence dependent rise of the 2PPE signal was found that could potentially be explained by dynamic changes in surface band alignment and the space charge region. However, due to this number of potential limitations great care are must be taken to draw clear conclusions and we decided to not address this topic in more detail without further investigation.

We agree with the reviewer's opinion that solely the 2PPE results do not provide an unambiguous picture of the carrier recombination. For this reason, we complemented the surfaces sensitive method with the bulk information we gained by photoluminescence measurements. As stationary PL reveals the main recombination channel occurs via defect-mediated radiative recombination originating from V_{Cu} vacancies. We could further show that any surface modification via Pt-deposition does not change the signal intensity and the ratio between the free exciton signal and defect PL. In contrast the Pt-deposition modified the surface lifetimes of both species, conduction band and defect band electrons. From both findings we draw the conclusion that the characteristic signal rise in the band-gap feature arises from carrier diffusion to the surface in the defect band. In response to this point we shifted and extended the discussion about the PL findings and their correlation with the 2PPE results.

Associated changes to the main text:

Page 8:

“In order to gain additional information about the origin of the filling of these defect levels located within the band gap photoluminescence (PL) measurements were conducted exhibiting primarily bulk sensitivity and, therefore, represent a well-suited complementary method to the surface sensitive 2PPE technique. The experiments were conducted at the same stages of sample preparation (reconstructed, Pt-deposited) without breaking UHV conditions and applying identical pump laser pulse conditions as utilized in the 2PPE measurements (Supplementary section 5). Both stationary and time-resolved photoluminescence measurements (PL) strongly suggest that Cu vacancies act as the dominant defect type in the investigated material, and the photoemission energy is in close accord with the energetic position of the defect state in the band gap. The different surface treatments did not alter the PL signals, suggesting a relatively high concentration of Cu vacancies in the bulk and that the surfaces contribution as well as influence to the charge recombination is negligible. We point out that the found energetic match does not allow to draw direct conclusions about the type and density of defects at the surfaces. Therefore, additional surface-sensitive experiments (LEED, XPS) were performed to obtain information about the stoichiometry, structure, and energetics (band bending, band alignment) of the material (see Supplementary section 6).

The main conclusion is that photoelectrons relax to bulk defect states prior to reaching surfaces and thus loose their potential very fast interior to the material. Does this mean that Cu₂O cannot be fixed? If the defect are due to Cu vacancies is there a way to decrease their number and see how that impacts the results.

The outcome of the study highlights the importance of bulk recombination processes in Cu₂O. Hence, in addition to known issues in Cu₂O, such as a mismatch between the electronic band alignment at junctions and defect states at interfaces, bulk defects in the Cu₂O may substantially limit the obtainable voltage in Cu₂O devices. High quality Cu₂O sheets or nano assemblies are therefore required to obtain efficient photocathodes. In recent studies (Refs.46,49,50) there have been significantly increased conversion efficiencies reported, however, without a clear correlation of these improvements with specific bulk properties. The manuscript therefore underlines the importance of material development in Cu₂O electrodes. We discuss these potential approaches in the main text but further analysis of their suitability and impact on the presented results is beyond the scope of the study.

Overall. I like the manuscript - but not sure that this should be published in Nature communication. The impact of the manuscript seems rather low. Maybe the authors should use the results to improve the operation of the photoelectrons.

We thank the reviewer for the comments and critical assessment of the work. We have addressed the concerns in the modifications shown above.

Reviewer #2 (Remarks to the Author):

In the manuscript 'Femtosecond time-resolved two-photon photoemission studies of ultrafast carrier relaxation in Cu₂O photoelectrodes' M. Borgwardt et al. report on a time-resolved 2PPE study of the carrier dynamics in a pristine Cu₂O single crystal and after deposition of a small amount of Pt. From their data they conclude on Cu bulk vacancy states acting as decay channels for charge carriers photoexcited into the conduction band and being responsible for the reduced photovoltages observed for this material system. The experiments have been performed thoroughly and the data were analyzed carefully. The main text itself misses sometimes relevant details, which are partly 'hidden' in the supplementary information. The interpretation stays at qualitative level but is convincing at least regarding the major conclusions.

As the key outcome of the study the authors conclude that the 'modest photovoltages that can be obtained from these Cu₂O samples are not primarily due to recombination induced by surface states, but primarily from bulk recombination processes into bulk defect states.' This seems to me an interesting finding, but I am not really convinced that this is enough to justify publication in Nature Communications. I have to admit, however, that experts in the field of solar-driven water splitting processes may come to a different conclusion.

In case the manuscript is considered for publication in Nature communications, the following points should be answered by the authors and considered in a revised version:

We thank the reviewer for carefully reading and evaluating the manuscript. We agree with the reviewer's opinion in that the manuscript needs a clearer focus on its central outcome and highlighting of its impact on future Cu₂O material development. We have adapted the manuscript accordingly (see changes to abstract and in response to the comments hereinafter). Below is our point by point response to all comments.

(1) The authors state in their conclusions that their 'observations indicate that tr-2PPE can provide a powerful experimental method to simultaneously unravel the energetics and dynamics of photoexcited electrons as they arrive at surfaces or internal interfaces of semiconductors. Tr-2PPE thus provides explicit insight into the origin and mechanism of photovoltage losses in photovoltaic or photocatalytic materials and can offer crucial insights at an early stage of material development.' This is definitely correct, but in my opinion not actually new or surprising. Tr-2PPE experiments have been performed over more than 30 years and a multitude of experiments on quite different material systems from many groups were published in the past, also including experiments addressing relevant processes in photovoltaic systems. I actually was very surprised to see that the authors refer only to a very limited number of tr2PPE studies (in the supplemental information, only). Even more, these citations are in my opinion not really of relevance for the key outcome of the study. I strongly recommend checking for instance tr2PPE studies by the Zhu group on inorganic-organic perovskites. As far as I know also the Perfetti group and the Fauster group (D. Niesner) have published some work on this

stuff. In a wider context also work by the Wolf group on charge carrier solvation in ice overlayers and work by the Harris group and the Höfer group on charge-transfer processes across interfaces could be of interest. Finally, I remember some work by the Aeschlimann group on surface charge accumulation in GaAs due to carrier diffusion from the bulk. I am pretty sure that at least part of this work could be helpful in the interpretation of the submitted study.

We thank the reviewer for pointing us to these well-known and highly relevant studies. We have included the suggested references in the manuscript.

Associated changes to the main text:

Page 3:

“2PPE spectroscopy has been previously successfully applied to study electron dynamics at surfaces and charge transfer processes across interfaces of metals and semiconductor model systems²²⁻²⁴. Only recently has its capabilities been utilized to study processes in emerging photovoltaic materials systems such as hot electron relaxation dynamics in hybrid metal-organic perovskite semiconductors²⁵⁻²⁷. We have now extended these efforts to the group of metal oxide semiconductors by including Cu₂O – one of the most promising metal oxide candidates for solar water splitting²⁸.”

Page 10:

“Similar findings have been reported for tr-2PPE studies on GaAs (100) surfaces where a rising electron population on a ps timescale has been assigned to scattering of electrons into low-energy states in the band-bending region³⁴.”

(2) It becomes not clear what amount of Pt was finally deposited onto the reconstructed Cu₂O surface that was investigated by tr2PPE. Is the surface fully covered with Pt or are there open Cu₂O areas in between the Pt islands that form upon deposition as verified by AFM and SEM. In the main text, the authors refer to a non-conformal structure of the Pt adlayer, which may hint to open Cu₂O areas. However, in the supplementary information (section 6) the authors write: ‘X-ray photoelectron spectroscopy (XPS) data for Pt films of increasing thickness revealed a non-conformal coverage for the thinnest films (0.5 nm) while a continuous Pt layer was obtained for longer deposition times (Fig. S8b). Based on these results, 1 nm Pt films were used for all of experiments discussed herein.’ What does this finally mean for the samples discussed in the main text? The information is in my opinion important as the presence of open Cu₂O areas could be of relevance for the interpretation of the data.

We thank the reviewer for pointing this out. Our XPS data and its theoretical modeling indicated a non-conformal growth for Pt deposition via UHV evaporation of films with

thicknesses < 1 nm. For films with theoretical thickness equivalent to 1 nm and larger – as determined by monitoring the rate of deposition using a quartz crystal microbalance (QCM) in UHV – the fitting procedure of the XPS data revealed a decrease in the size effects (line shape asymmetry, shift of the Pt 4f binding energy, linewidth broadening) associated with non-conformal growth. However, AFM and SEM verified the non-conformal growth of Pt for the 1 nm thick layers as used in the samples for tr-2PPE measurements.

Associated changes to the Supplementary text:

Page 11:

“To summarize, X-ray photoelectron spectroscopy (XPS) data and its theoretical modeling indicated a non-conformal growth for Pt deposition via UHV evaporation of films with thicknesses < 1 nm. For films with theoretical thickness equivalent to 1 nm and larger – as determined by monitoring the rate of deposition using a quartz crystal microbalance (QCM) in UHV – the fitting procedure of the XPS data revealed a decrease in the size effects (line shape asymmetry, shift of the Pt 4f binding energy, linewidth broadening) associated with non-conformal growth (Fig. S8b). Based on these results, Pt films with QCM thickness equivalent to 1 nm were used for all of the experiments discussed herein.”

(3) The authors directly conclude from the bulk-sensitive photoluminescence measurements onto the origin of the dominating signal recorded using the surface sensitive 2PPE method. I would suggest being more conservative with such statements as it is well known that the surface electronic structure can differ significantly from the bulk electronic structure as the bulk termination can result in a surface reconstruction and also the formation of quite different types of defects.

We thank the reviewer for this valuable comment. We agree that the current version of the manuscript reads as it was our intention to draw conclusions about the actual type and density of the surface defects from photoluminescence measurements. We point out that their energetic position in the bulk (measured by PL) and at the surface (tr-2PPE) match and conclude that “surface accumulation of the electrons occurs at levels isoenergetic with the bulk trap band“. We are aware of the fact that the surface reconstruction and adsorption of adlayers or atoms can decisively alter the surface properties compared to the bulk. We clarified the text regarding this point.

Associated changes to the main text:

Page 8:

“In order to gain additional information about the origin of the filling of these defect levels located within the band gap photoluminescence (PL) measurements were conducted exhibiting primarily bulk sensitivity and, therefore, represent a well-suited complementary method to the surface sensitive 2PPE technique. The experiments were conducted at the same stages of sample preparation (reconstructed, Pt-deposited)

without breaking UHV conditions and applying identical pump laser pulse conditions as utilized in the 2PPE measurements (Supplementary section 5). Both stationary and time-resolved photoluminescence measurements (PL) strongly suggest that Cu vacancies act as the dominant defect type in the investigated material, and the photoemission energy is in close accord with the energetic position of the defect state in the band gap. The different surface treatments did not alter the PL signals, suggesting a relatively high concentration of Cu vacancies in the bulk and that the surfaces contribution as well as influence to the charge recombination is negligible. We point out that the found energetic match does not allow to draw direct conclusions about the type and density of defects at the surfaces. Therefore, additional surface-sensitive experiments (LEED, XPS) were performed to obtain information about the stoichiometry, structure, and energetics (band bending, band alignment) of the material (see Supplementary section 6).“

(4) In line 112 the authors state that the Cu₂O-derived features on the Pt covered sample primarily originate from a combination of directly emitted electrons from the Cu₂O substrate (I assume this refers to Cu₂O areas not covered by Pt) in conjunction with electrons probed through the Pt top layer. I have to admit that I do not completely understand how the authors come to this conclusion from their discussion on the electron mean free path in the same paragraph. Particularly I am wondering how the authors can exclude that the Cu₂O-derived features on the Pt covered sample exclusively arise from directly emitted electrons from the Cu₂O substrate?

We agree with the reviewer that this point needs to be clarified. Given that the diameter of the probe laser spot was on the order of 50µm, it implies that the probed area of the Pt covered sample contained both Pt islands and uncovered Cu₂O areas. Hence, we obtained an averaged 2PPE signal containing both contributions. In addition, since the probing depth of the 2PPE measurement exceeds the thickness of the non-conformal Pt layer, the signal contains both electrons probed through the Pt and as well as emitted from the uncovered Cu₂O substrate. We have modified the corresponding section in the manuscript to extend and clarify this point.

Associated changes to the main text:

Page 7:

“For low photoelectron kinetic energies, the inelastic mean free paths are relatively large, exceeding tens of nanometers²⁹. However, because elastic electron-acoustic phonon scattering overcomes the energy loss scattering events, the photoelectron escape depth is substantially reduced for such low kinetic energies^{30,31}. Hence, the thickness region sampled by 2PPE is typically on the order of a few nanometers^{25,32,33}. In addition to the above mentioned surface sensitivity of 2PPE, it is important to note that due to the non-conformal coverage of the Pt adlayer and the surface area accessible by the probe pulse with diameter of ~50 µm both Pt covered and uncovered Cu₂O areas were simultaneously sampled. We therefore expect that the Cu₂O-derived features on the Pt covered sample primarily originate from a combination of directly

emitted electrons from the Cu₂O substrate in conjunction with electrons probed through the Pt top layer.”

(5) The conclusion that the characteristic signal rise in the band-gap feature arises from carrier diffusion to the surface in the defect band seems to me very reasonable. I was surprised that the observed timescales were not related to the time-resolved PL data on the (much longer) defect depopulation times. Furthermore, the authors observe and mention in the main text a two-component signal rise with a pronounced fluence dependence of the fast rise-time. What can we learn from these observations? This point is not discussed at all in the rest of the manuscript.

We agree that a direct correlation of the time scales between the two methods seems an attractive opportunity to learn more about the physics involved. However, we believe that the very different time resolutions, recorded time windows and information depth of both measurements complicate their interpretation and comparison with regard to dynamics. Although we see a decrease in the tr-2PPE defect signal within the available 2ns time window, that might be assigned to a (bulk) defect depopulation, it is not clear how the signal further evolves in time and how fast it reaches zero level or whether it converges into a long-lived offset. On the other hand, the defect related PL signal reaches its maximum at about 10ns. We further point to the fact that the 2PPE signal is particularly sensitive to the charge carrier dynamics in the space-charge region that might considerably deviate from the bulk, especially with regard of the initial diffusion process of charge carriers into the surface trap states. Dynamic changes in surface band alignment and the space charge region induced by the optical pump could also explain the fluence dependent rise of the 2PPE signal. Due to this number of potential limitations great care must be taken to draw clear conclusions and we decided to not address this topic in more detail without further investigation.

(6) What mechanism channels excited electrons at the surface into sites associated with a high dislocation density (line 214)? Shouldn't such type of process result in the formation of an in plane surface potential gradient which at some point will stop the channeling process.

The diffusion process towards the surface is driven by the downward band bending. Due to the non-uniform defect distribution at the reconstructed surface charge carrier accumulation will be statistically enhanced at sites associated with a higher dislocation density. We agree that the current version of the manuscript seems to suggest a specific effect being the origin for the charge separation. We also agree that a non-uniform charge distribution at the surface would ultimately lead to potential gradients that will counteract the diffusion process. However, as pointed out answering question 4

the limited spatial resolution in the tr-2PPE as well as XPS measurements do not allow to draw any conclusion on this level of detail.

Associated changes to the main text:

Page 12:

“The long rise time of the integrated 2PPE signal, which extended into the ns time domain, is consistent with these states being filled with bulk electrons that drift or diffuse toward the surface with the electrons then primarily accumulated at sites with a high dislocation density.”

(7) The author mention thermally assisted hopping within the V_{Cu} defect band. What energies are we talking about? Here, experiments performed at different temperatures would be nice and would strongly support the interpretation in terms of bulk to surface diffusion.

We thank the reviewer for this comment and have added a reference that emphasizes low electron mobility in Cu₂O using terahertz spectroscopy implying non-conduction band and activated transport. Considering the energetic position of the defect band electrons cannot repopulate the conduction band and must hop between defect levels. To go into full details of the bulk transport mechanism temperature dependent experiments would be nice but would go beyond the scope of this manuscript.

Associated changes to the main text:

Page 12:

“This fit well to the low electron mobility estimated by Paracchino et al.³⁹ between 2.7–6.3 cm²V⁻¹s⁻¹ in electrodeposited Cu₂O using time-resolved terahertz spectroscopy. Considering the position of the V_{Cu} defect band located deep below the conduction band minimum bulk electron transport to the surface would consistently occur via thermally assisted hopping within the defect band, in accord with the moderate values for minority-carrier mobility and diffusion length from other reports for Cu₂O^{11,40–42}”

(8) In the abstract the authors claim that their data suggest that the Pt adlayer reduced the surface defect states. Later in the manuscript (line 147) they write that ‘...either the defect states disappeared after deposition of Pt, or that the Pt provided an alternative pathway for the photoexcited electrons that prevented filling of defect states.’ From the following discussion in the manuscript I learned that none of these statements is finally correct: The defects did not disappear but become hidden underneath the Pt-island. These defect states become still populated (at least as implied by Fig. 4(b)). However,

in the presence of the Pt overlayer they get much faster depopulated in comparison to the reconstructed surface due to coupling to platinum electronic states.

We agree that the abstract needs clarification regarding the final conclusion made in the manuscript. The preliminary interpretation in the result section (Page 9) considering solely the tr-2PPE signals states two different possibilities, that are either a reduction of available surface trap state or an alternative relaxation channel that prevents the accumulation of charge in the defect state. After incorporating of all other available information, we concluded that the latter is the most probable explanation for the missing signal rise. We adapted the manuscript at both places in the text.

Associated changes to the abstract:

“The data also suggest that the Pt adlayer suppresses the slow accumulation of electrons into the surface defect states and that the Pt is capable of mediating the charge transfer at the semiconductor/metal interface.”

Associated changes to the main text:

Page 9:

“This behavior suggests that either the defect states disappeared after deposition of Pt, or that the Pt provided an alternative pathway for the photoexcited electrons that prevented accumulation of charge carriers in the defect states.”

(9) The authors state that the reconstructed surface has a non-uniform defect distribution (line 143) (and conclude from this that the ‘long-lived signal can consistently be ascribed to sites associated with a high dislocation density.’) I missed somehow, where this information is coming from. Maybe this point is somewhere hidden in the supplementary information. In any case at least a reference must be given.

Results obtained by AFM and SEM indicate a non-uniform defect distribution (both presented in the supplementary information). We agree that there is no reference in the main text to this important information. We therefore adapted the main text accordingly.

Associated changes to the main text:

Page 9:

"Surface analysis of the reconstructed and Pt covered Cu₂O sample by means of atomic-force and scanning electron microscopy (AFM, SEM; see Supplementary section 8) revealed a non-uniform defect distribution, hence the long-lived signal can consistently be ascribed to sites associated with a high dislocation density."

(10) *It seems to be very important that a reconstructed Cu₂O surface instead of a non-reconstructed surface is used in the experiment as this point is mentioned several times throughout the manuscript. I am not familiar with the details of the water-splitting capabilities of Cu₂O and I think it would be helpful for the non-expert to spend a few words in the main text or the supplementary information on this point.*

We agree with the reviewer that this point needs clarification: all tr-2PPE, PL and tr-PL measurements were also conducted directly after crystal cleavage without further surface treatment (as-received) resulting into qualitatively similar signals compared to the reconstructed samples. However, signal intensities and background contributions with tr-2PPE were different and varied depending on respective sample and chosen spot on the sample surface. Such variations are not surprising due to the high surface sensitivity, so that we decided to include only the results from the well-defined and highly reproducible surface that were not hampered by these difficulties. The crystal orientation and reconstruction procedure were not specifically chosen with regard to water-splitting aspects of Cu₂O and we will add this information to the manuscript.

Associated changes to the main text:

Page 4:

“Artificially synthesized Cu₂O crystals were chosen and an extensive surface reconstruction procedure was employed that ensures high reproducibility accounting for the surface sensitivity of the tr-2PPE technique. Investigations conducted at crystals without any surface treatment (as-received) yielded qualitatively similar results and are not shown.”

Reviewer #3 (Remarks to the Author):

This work measures the relaxation dynamics of photoexcited electrons in Cu₂O pure and with a Pt cover. The performed 2-photo-photoelectron measurements are well designed and executed. The manuscript describes the undertaken experiments and analyses clearly. I also summarizes the results clearly. The difference between pure and Pt-covered material is strikingly different and I think that the authors are right with their analysis. The defect sites on the surface become slowly filled with bulk electrons. it is quite fascinating to see this in real time.

I only have one very minor comment: in the abstract the authors write about energetically "unfavorable" states. It is not upfront clear what that should mean. The authors may want to chose a more clarifying description.

We thank the reviewer for studying our manuscript and for providing positive feedback and comments. We have rewritten the abstract and replaced the term “unfavorable” accordingly.

REVIEWERS' COMMENTS:

Reviewer #1 (Remarks to the Author):

The authors have adequately addressed this reviewers comments. I can recommend publication.

Reviewer #2 (Remarks to the Author):

I carefully read the reply of the authors to the comments of all three reviewers. TIn my opinion, the authors addressed all points raised by the reviewers in a satisfactory manner. I particularly acknowledge that the revisions made by the author clarified the actual focus of the study. Overall I can agree with a publication of the manuscript. However, as already mentioned in my first review, I am not an expert in the field of solar-driven water splitting and have therefore difficulties to judge whether the novelty or relevance of the outcome of the study justifies publication in Nature Communications.

Point-by-point response to the referees' comments:

Reviewer #1 (Remarks to the Author):

The authors have adequately addressed this reviewers comments. I can recommend publication.

We thank the reviewer once again and gratefully acknowledge the recommendation to publish.

Reviewer #2 (Remarks to the Author):

I carefully read the reply of the authors to the comments of all three reviewers. TIn my opinion, the authors addressed all points raised by the reviewers in a satisfactory manner. I particularly acknowledge that the revisions made by the author clarified the actual focus of the study. Overall I can agree with a publication of the manuscript. However, as already mentioned in my first review, I am not an expert in the field of solar-driven water splitting and have therefore difficulties to judge whether the novelty or relevance of the outcome of the study justifies publication in Nature Communications.

We thank the reviewer once again and gratefully acknowledge the recommendation to publish.